# Sudapyridine (WX-081), a Novel Compound against *Mycobacterium tuberculosis*

Rong Yao,[a,b] Bin Wang,[a,b] Lei Fu,[a,b] Lei Li,[c] Kejun You,[c] Yong-Guo Li,[c] (iD) Yu Lu[a,b]

aBeijing Key Laboratory of Drug Resistance Tuberculosis Research, Beijing Tuberculosis and Thoracic Tumor Research Institute, Beijing, China
bBeijing Chest Hospital, Capital Medical University, Beijing, China
cShanghai Jiatan Biotech Ltd., a subsidiary of Guangzhou JOYO Pharma Ltd., Shanghai, China

Rong Yao and Bin Wang contributed equally to the article. Author order was determined by the corresponding author after negotiation.

**ABSTRACT** Bedaquiline (BDQ) was historically listed by the World Health Organization (WHO) in 2018 as the preferred option for rifampin-resistant tuberculosis (RR-TB) and multidrug-resistant tuberculosis (MDR-TB). However, when there is no other effective regimen, the side effects and weaknesses of BDQ limit its use of MDR-TB. There is a black box warning in the package insert of BDQ to warn patients and health care professionals that this drug may increase the risk of unexplained mortality and QT prolongation, which may lead to abnormal and potentially fatal cardiac rhythm. In addition, the phenomenon of elevated liver enzymes in clinical trials of BDQ is a potential sign of hepatotoxicity. Therefore, it is still a medical need to develop new compounds with better safety profiles, patient compliance, affordability, and the ability to retain the efficacy of BDQ. After extensive lead generation and optimization, a new analog, sudapyridine (WX-081), was selected as a potential new antituberculosis candidate to move into clinical trials. Here, we evaluated WX-081's overall preclinical profile, including efficacy, pharmacokinetics, and toxicology. The *in vitro* activity of WX-081 against drug-sensitive and drug-resistant tuberculosis was comparable to that of BDQ, and there was comparable efficacy between WX-081 and BDQ in both acute and chronic mouse tuberculosis models using low-dose aerosol infection. Moreover, WX-081 improved pharmacokinetic parameters and, more importantly, had no adverse effects on blood pressure, heart rate, or qualitative ECG parameters from nonclinical toxicology studies. WX-081 is under investigation in a phase 2 study in patients.

**IMPORTANCE** This study is aimed at chemotherapy for multidrug-resistant tuberculosis (MDR-TB), mainly to develop new anti-TB drugs to kill *Mycobacterium tuberculosis*, a microorganism with strong drug resistance. In this study, the structure of a potent antituberculosis compound, bedaquiline (BDQ), was optimized to generate a new compound, sudapyridine (WX-081). This experiment showed that its efficacy was similar to that of BDQ, its cardiotoxicity was lower, and it had good kinetic characteristics. This compound will certainly achieve significant results in the control and treatment of tuberculosis in the future.

**KEYWORDS** WX-081, anti-TB drug, *Mycobacterium tuberculosis*

Tuberculosis is a type of chronic communicable disease resulting from *Mycobacterium tuberculosis* (MTB), which can affect multiple organ systems. The increased number of MDR-TB and extensively drug-resistant tuberculosis (XDR-TB) has posed a great challenge to the treatment and control of tuberculosis. As mentioned by the Global Alliance for TB Drug Development, new agents, shortening or simplifying the effective treatment of TB, would greatly improve TB control programs (1). BDQ, the first new antituberculosis drug on the market in 50 years, is active against *M. tuberculosis in vivo* and *in vitro*, including sensitive

Address correspondence to Yu Lu, luyu4876@hotmail.com, or Yong-Guo Li, ygli@joyopharma.com.

The authors declare no conflict of interest.

**FIG 1** Structure of WX-081.

strains, multidrug-resistant strains, and dormant bacteria, with MIC ranging from 0.03 to 0.12 $\mu$g/mL (2, 3). It exerts bacteriostatic and bactericidal effects by inhibiting the ATP (ATP) synthase of MTB (2, 4, 5). BDQ is a useful drug in regimens for the treatment of drug-resistant TB in humans, however, its use has been partially limited by toxicity issues, which include QT prolongation, hepatotoxicity, and phospholipidosis (6–10). In addition, the elimination half-life of BDQ is 5.5 months (11), which may be due to its slow release from peripheral tissue compartments (11), resulting in a complicated dose design in clinical for the inconvenience of patient compliance (12). Chemically, BDQ is expensive with a high cost of goods and cost burden for the patients and TB control. Thus, it is particularly important to rationally modify and optimize the structure of BDQ to discover a better compound.

We reported on the preclinical development of sudapyridine (WX-081), a novel clinical candidate for the therapy of MTB, including antimycobacterial properties and pharmacokinetic as well as toxicology. Its clinical potential is currently being evaluated in ongoing clinical trials in patients (13).

## RESULTS

**Chemistry and *in vitro* antimycobacterial activity.** The most active compound of the class, WX-081 (Fig. 1), modified the quinoline group based on retaining the pharmacological active site of BDQ (consisting of a quinolinic central heterocyclic nucleus and side chains of tertiary alcohol and tertiary amine groups) to transform it into a pyridine group.

WX-081 has a unique spectrum of strong antimycobacterial activity *in vitro* (Table 1). The MICs range of the international reference strain *M. tuberculosis* H37Rv and five antibiotic-sensitive isolates was 0.117 to 0.219 $\mu$g/mL (Table 1) versus <0.02 to 0.034 $\mu$g/mL for isoniazid and 0.022 to 0.066 $\mu$g/mL for rifampin. WX-081 demonstrated similar or minor weakness *in vitro* activity against TB drug-resistant clinical isolates of *M. tuberculosis*, including isoniazid, rifampin, streptomycin, ethambutol, and moxifloxacin. WX-081 retained activity against MDR-TB strains. All 15 isolates of MDR-TB tested were susceptible to WX-081 at <1 $\mu$g/mL (Table 1). The results showed that WX-081 had good *in vitro* activity against clinically isolated strains of *Mycobacterium tuberculosis*, including sensitive strains and drug-resistant strains. The activity is similar to that of BDQ, and there was no cross-resistance with other antituberculosis drugs.

**MBC determination.** The minimum bactericidal concentration (MBC) of WX-081 or BDQ was determined at the concentration of their MIC to be 16× MIC against *M. tuberculosis* H37Rv and 5 clinical isolates. Based on these results, the MBC of WX-081 and BDQ against *M. tuberculosis* was not greater than 32× MIC (Table 2). Therefore, similar to that of BDQ, WX-081 should have bactericidal activity.

**Time-kill curve analysis.** The time-killing test results of WX-081 and BDQ against *M. tuberculosis* H37Rv, drug-susceptible, and MDR-TB strains are in Fig. 2. WX-081 displayed bactericidal effects on H37Rv and two clinical isolates at 4× MIC and above. The results were consistent with those of MBC.

**Pharmacokinetics (PK).** A head-to-head comparison of WX-081 and BDQ in mice and rats was performed in addition to the dog PK study. The pharmacokinetic parameters are provided in Table 3. The results showed better exposure of WX-081 than BDQ in mice and rats. Significantly, the exposure of the target organ in the lung at 96 h of WX-081 was several times that of BDQ. After oral administration of WX-081 at doses of

**TABLE 1** MICs of WX-081 against different mycobacterial species[a]

| Clinical strains | MICs (ug/mL) | | | | | |
|---|---|---|---|---|---|---|
| | WX-081 | BDQ | INH | RFP | MFX | KM |
| 13946: resistant to INH SM RFP EMB RBT PAS OFX | 0.098 | 0.021 | 1.813 | >40 | 0.220 | 0.788 |
| 14550: resistant to INH SM RFP EMB KM CPM AK 1321 PAS | 0.170 | 0.051 | 2.251 | >40 | 1.129 | 1.186 |
| 14822: resistant to INH SM RFP EMB KM RBT AK 1321 PAS | 0.480 | 0.111 | >40 | >40 | 1.001 | >40 |
| 14158: resistant to INH SM RFP EMB AK 1321 PAS OFX | 0.111 | 0.030 | >40 | >40 | 2.329 | >40 |
| 14441: resistant to INH RFP RBT AK CPM 1321 OFX LFX | 0.465 | 0.214 | >40 | >40 | 2.122 | >40 |
| 14470: resistant to INH SM RFP EMB KM RBT AK CPM 1321 PAS | 0.966 | 0.468 | >40 | >40 | 4.633 | 0.437 |
| 13908: resistant to INH RFP EMB 1321 PAS OFX LFX | 0.849 | 0.266 | 3.008 | 6.904 | 1.742 | 1.151 |
| 14282: resistant to INH SM RFP EMB RBT 1321 OFX LFX | 0.113 | 0.053 | 2.347 | >40 | 0.307 | 1.199 |
| 15321: resistant to INH RFP EMB AK ETO OFX RFT | 0.109 | 0.028 | 1.037 | >40 | 0.384 | >40 |
| 14619: resistant to INH RFP KM CPM AK 1321 PAS | 0.300 | 0.052 | 4.377 | >40 | 1.098 | 1.195 |
| 16030: resistant to INH RFP RFT | 0.057 | 0.017 | 2.448 | >40 | 0.019 | 1.072 |
| 14862: resistant to INH SM RFP EMB 1321 CPM PAS | 0.026 | 0.006 | 2.166 | 15.216 | 1.053 | 1.243 |
| 16825: resistant to INH RFP AK RFT | 0.226 | 0.060 | 2.373 | >40 | 0.501 | >40 |
| 15833: resistant to INH RFP RFT | 0.236 | 0.090 | 4.495 | >40 | 0.066 | 2.302 |
| 14092: resistant to INH SM RFP | 0.235 | 0.048 | 2.406 | >40 | 0.605 | 1.199 |
| Susceptible strain 14241 | 0.117 | 0.029 | 0.034 | 0.035 | 0.122 | 2.417 |
| Susceptible strain 14242 | 0.219 | 0.031 | 0.028 | 0.066 | 0.037 | 1.146 |
| Susceptible strain 14280 | 0.124 | 0.053 | 0.024 | 0.057 | 0.035 | 1.219 |
| Susceptible strain 14286 | 0.152 | 0.043 | 0.030 | <0.02 | 0.039 | 1.154 |
| Susceptible strain 14480 | 0.120 | 0.037 | <0.02 | 0.046 | 0.038 | 1.907 |
| H37Rv | 0.070 | 0.012 | 0.031 | 0.014 | 0.045 | 0.730 |

[a]Ethambutol, EMB; isoniazid, INH; rifampicin, RFP; rifapentine, RFT; rifabutin, RBT; ethionamide, ETO; streptomycin, SM; capreomycin, CPM; para aminosalicylic acid, PAS; protionamide, Th1321; levofloxacin, LFX; moxifloxacin, MFX; ofloxacin, OFX; kanamycin, KM; amikacin, AK; bedaquiline, BDQ.

2, 6, or 20 mg/kg in beagle dogs, the time to peak concentration ($T_{max}$) was 4.6 to 7.7 h, and the plasma half-life ($t_{1/2}$) was 51 to 58h. Maximum drug concentration ($C_{max}$) and concentration-time area under the curve ($AUC_{0-inf}$) increased with dose, $C_{max}$ ranged from 390 to 1660 ng/mL, $AUC_{0-inf}$ ranged from 9490 to 58200 ng × h/m. There was no significant difference in the plasma half-life between oral and intravenous administration of WX-081 in beagle dogs. Intravenous injection of each different single dose of WX-081 allowed assessment of the volume of distribution and clearance across species.

**In vivo efficacy in mice.** Before the in vivo activity test, a preliminary in vivo toxicity study was conducted. All mice tolerated WX-081 at the administered doses. No adverse events were found. The dose that caused at least 50% of mice to die ($LD_{50}$) exceeded 3000 mg/kg.

In the acute infection model, the mean ($\pm$ standard deviation [SD]) pulmonary bacterial burden of the untreated control group increased from 4.39 $\pm$ 0.03 $\log_{10}$ colony forming unit (CFU) to 4.97 $\pm$ 0.08 $\log_{10}$ CFU during the 20-day course of chemotherapy. At the end of the treatment period, the average lung CFU counts of WX-081 treatment groups with 10 mg/kg and 20 mg/kg doses were 2.2 to 3.2 log units lower than those of the untreated control group ($P < 0.05$) (Fig. 3). WX-081 exhibited dose-dependent antituberculosis activity at doses ranging from 5 to 20 mg/kg. The activity of 20 mg/kg WX-081 had no difference from that of BDQ at

**TABLE 2** MBCs of WX-081 against different mycobacterial species

| Clinical strains | MBCs (ug/mL) | |
|---|---|---|
| | WX-081 | BDQ |
| 15321: resistant to INH RFP EMB AK ETO OFLX RFT | >1.25 | >0.313 |
| 14441: resistant to INH RFP RBT AK CPM 1321 OFLX LVFX | >5 | >2.5 |
| 14619: resistant to INH RFP KM CPM AK 1321 PAS | 0.313 | 0.156 |
| 15833: resistant to INH RFP RFT | >1.25 | >0.32 |
| Susceptible strain 14242 | 0.313 | 0.156 |
| H37Rv | 2.500 | 0.156 |

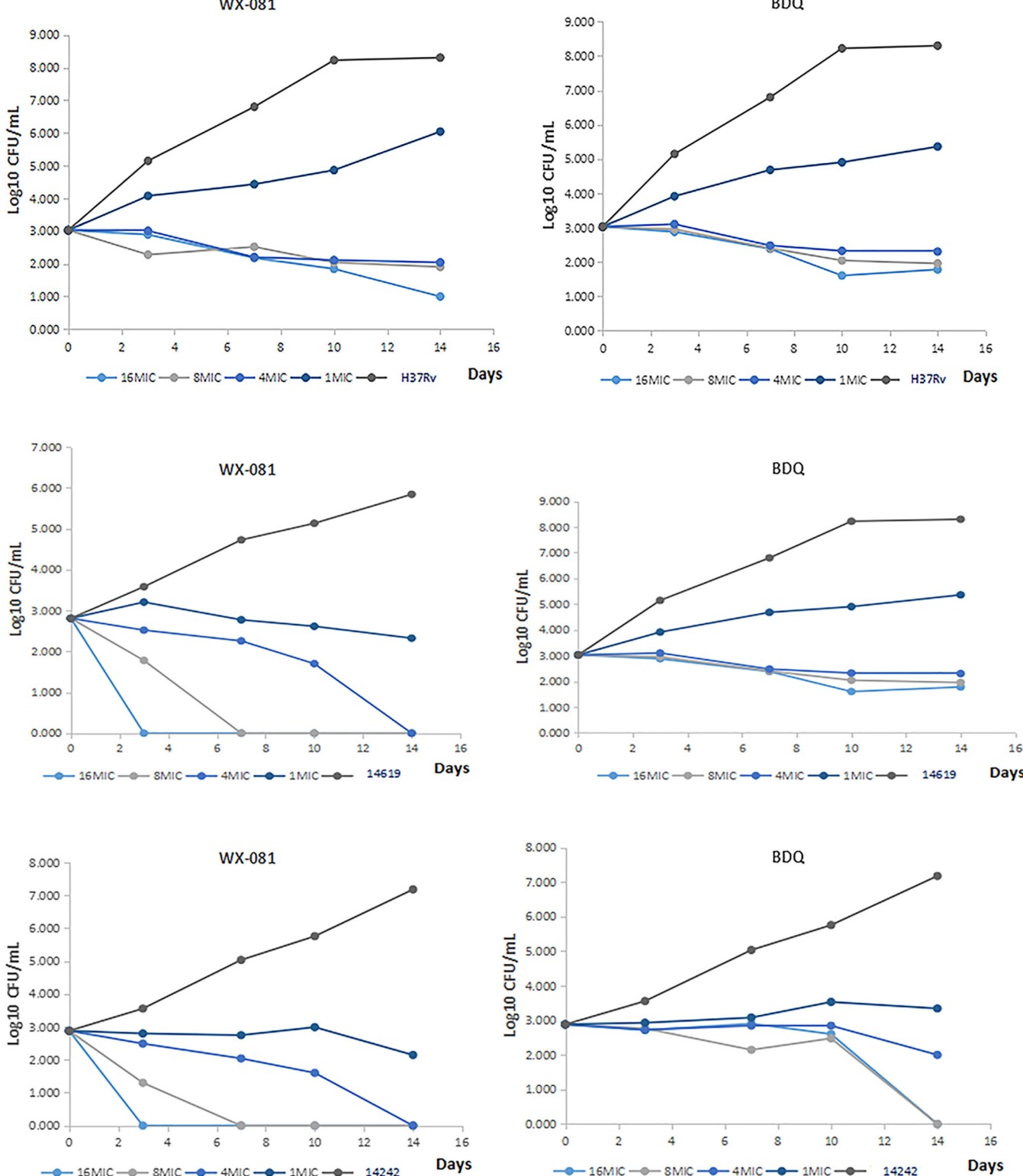

**FIG 2** The time-killing curve of WX-081 and BDQ against *M. tuberculosis* H37Rv and two clinical isolates. 14619: resistant to INH RFP KM CPM AK 1321 PAS; 14242: susceptible strain.

the equivalent dose.WX-081 at 10 mg/kg affected a larger reduction in CFU than BDQ at 10 mg/kg ($P < 0.05$).

Next, we evaluated the efficacy of WX-081 in mice chronically infected with low-dose aerosol *M. tuberculosis* H37Rv and determined the lung CFU after 8 weeks of

**TABLE 3** *In vivo* PK profiles of WX-081 and BDQ[a]

| Compound | Species | Dosage (mg/kg) | Vd$_{ss}$ (L/kg) | Cl (mL/min/kg) | t$_{1/2}$ (h) | Cmax (ng/mL) | T$_{max}$ (h) | AUC$_{0-last}$ (ng×h/m) | AUC$_{0-inf}$ (ng×h/m) | F (%) | At 96 h lung/plasma |
|---|---|---|---|---|---|---|---|---|---|---|---|
| BDQ | Mice | Iv. 1.0 | 6.21 | 7.76 | 21.3 | | | 2103 | 2158 | | 27.1/1.31 |
| | | Po. 6.25 | | | 47.6 | 608 | 5.33 | 6038 | 6358 | 47.1 | 135/4.56 |
| | Rat | Iv. 1.0 | 10.3 | 8.67 | 37.6 | to | | 1826 | 1931 | | 62.9/1.89 |
| | | Po. 5.0 | | | 29.3 | 354 | 0.83 | 2614 | 2677 | 27.7 | 58.8/1.52 |
| WX-081 | Mice | Iv. 1.0 | 10.4 | 3.59 | 46.3 | to | | 3991 | 4797 | | 241/12.4 |
| | | Po. 6.25 | | | 43.9 | 503 | 7.33 | 10155 | 11908 | 40.7 | 857/25.3 |
| | Rat | Iv. 1.0 | 9.2 | 8.25 | 25.6 | | | 2224 | 2351 | | 235/3.61 |
| | | Po. 5.0 | | | 29.8 | 328 | 1.00 | 4075 | 4312 | 36.7 | 337/4.74 |
| | Dog | Iv. 0.5 | 6.04 | 2.04 | 56.2 | | | 3610 | 4250 | | |
| | | Po. 2.0 | | | 51.1 | 390 | 4.67 | 8500 | 9490 | 58.9 | |
| | | Po. 6.0 | | | 55.6 | 1080 | 5.33 | 20200 | 22800 | 46.6 | |
| | | Po. 20.0 | | | 57.6 | 1660 | 7.67 | 50400 | 58200 | 34.9 | |

[a]V, volume of distribution; Cl, clearance; t$_{1/2}$, half-life; C$_{max}$, maximum drug concentration; T$_{max}$, peak time; AUC, concentration-time curve area; F, bioavailability; At 96 h lung/plasma, ratio of drug exposure of lung and plasma at 96 h. Both WX-081 and BDQ conducted PK studies in mice and rats, but only WX-081 was used for PK studies in dogs.

treatment. In this study, the lung CFU declined in mice treated for 8 weeks receiving WX-081 of any dose. Although the activity of WX-081 at 5, 10, and 20 mg/kg orally was slightly lower than BDQ, there was no significant difference in activity (Fig. 4). No toxicity was observed in any efficacy experiments through clinical observation and body weight monitoring (unpublished data).

**Toxicology.** In the head-to-head comparison study, oral administration of WX-081 and bedaquiline fumarate at 200 mg/kg once daily to male and female dogs for 14 days did not result in mortality and moribundity of animals (Table 4). Overall, the severity and incidence of changes of clinical signs, body weight, food consumption, echocardiogram (ECG), hematology, coagulation, serum chemistry, and changes in histopathology related to treatment with WX-081 were similar or less than that with bedaquiline fumarate in severity or incidence (unpublished data).

To evaluate the cardiac liability linking to QT elongation, four compounds, BDQ and its metabolite (BDQ-M2), WX-081, and its metabolite (WX-081-M3), were performed the *in vitro* assay of a human ether-á-go-go-related (*hERG*) ion channel. The results were obtained with the IC$_{50}$ of BDQ and WX-081 (IC50, BDQ >30.00 $\mu$M, WX-081 > 30.00 $\mu$M,

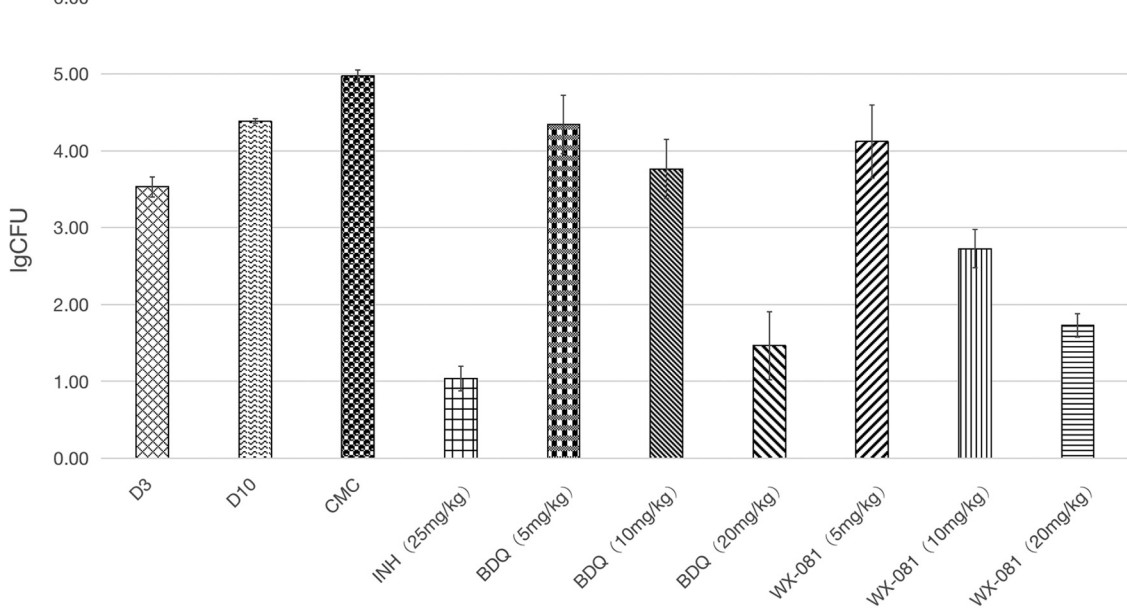

**FIG 3** Efficacy of WX-081in the acute mouse infection model. Dosages are in mg/kg. At the beginning of treatment at day 10 postinfection, there were 4.39 ± 0.03 log$_{10}$ CFU in the lungs of untreated mice. Values were determined after 20 days of treatment in BALB/c mice infected with *M. tuberculosis* H37Rv.

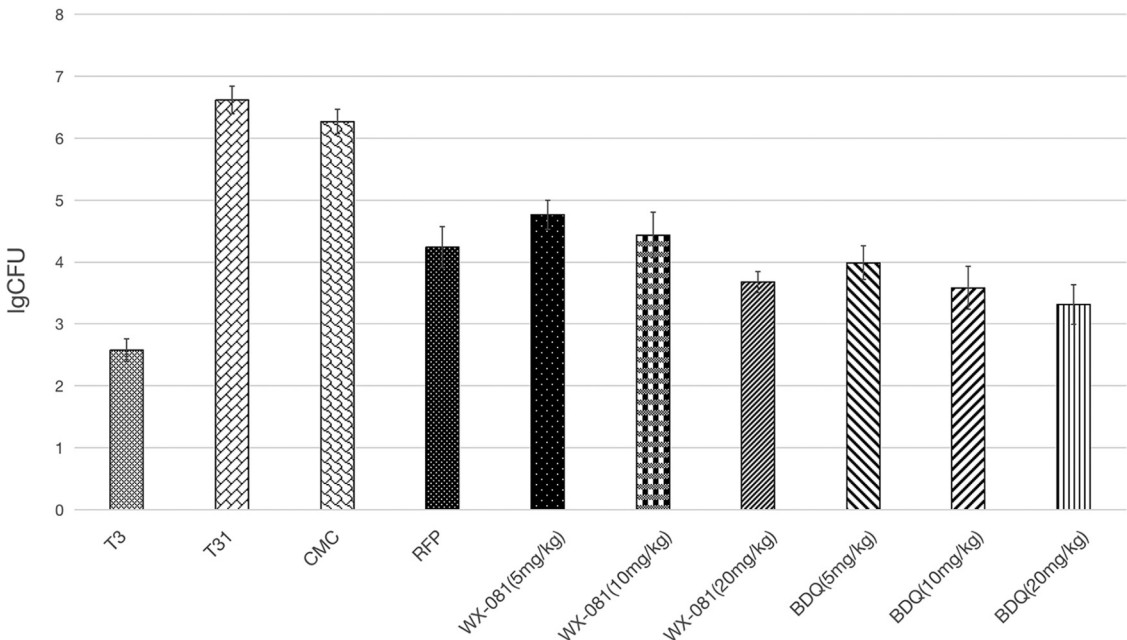

**FIG 4** Efficacy of WX-081 in the chronic mouse infection model. 8 weeks of treatment with vehicle (CMC), 10 mg/kg RFP, treatment initiated on day 28 postinfection. Dosages are in mg/kg. All mice were sacrificed before treatment or after 8 weeks of treatment.

respectively) with no concern of hERG and cardiac effects. The potential QT elongation concern was from its metabolite of WX-081 (IC$_{50}$, WX-081-M3 1.89 $\mu$M) and of BDQ (IC$_{50}$, BDQ-M2 1.73 $\mu$M). Reduced metabolite of WX-081 (WX-081-M3) was generated than BDQ-M2 in animals (Table 4) as well as in humans (unpublished data). There were no qualitative ECG changes associated with the oral administration of WX-081. On the contrary, the oral administration of bedaquiline fumarate may be associated with an increased incidence of first and second-degree atrioventricular (AV) block, which would be consistent with the evidence of PR interval prolongation noted to follow the administration of bedaquiline fumarate. There was also evidence of significant QTc interval prolongation.

## DISCUSSION

Due to the long treatment period of antituberculosis, large drug side effects, and poor patient adherence, more and more patients develop serious MDR-TB and extensively drug-resistant tuberculosis (XDR-TB) (14). Over the past 50 years, the United States Food and Drug Administration (FDA) has rarely approved new drugs for the treatment of tuberculosis, especially drug-resistant tuberculosis (DR-TB), which reflects the difficulties in the development of anti-TB drugs. Therefore, it is urgent to develop

**TABLE 4** Cardiotoxicity of WX-081 and bedaquiline in beagle dogs[a]

| Cardiac indicators | ECG parameters | WX-081 | Bedaquiline fumarate |
|---|---|---|---|
| hERG IC$_{50}$ (uM) | | >30.00 | >30.00 |
| IC$_{50}$ of metabolite | | 1.89 | 1.73 |
| AUC$_{0-24h}$ metabolite-to-parent ratio | | 0.039-0.44 | 0.082-0.60 |
| cTnI (pg/mL) | | | 440 (1/3 male) |
| ECG | HR | Slowing of the heart rate | An increased incidence of first- and second-degree atrioventricular block |
| | RR/PR interval | Lengthening of the RR interval | Prolongation of PR interval |
| | QT interval | Physiologically appropriate lengthening of the QT interval | Significant QT interval prolongation |

[a]Fourteen days repeated oral dose toxicity and toxicokinetic, head-to-head study in beagle dogs. hERG IC$_{50}$, half-maximum inhibitory concentration of the hERG channel; AUC$_{0-24h}$, time curve in a dosing interval of 24 h; cTnI, cardiac troponin I; ECG, electrocardiogram; HR, heart rate.

new and efficient anti-TB agents to achieve effective control of tuberculosis. The structural modification of existing anti-TB drugs or active substances to obtain anti-TB new drugs with shorter courses, higher efficacy, and less toxic side effects is undoubtedly one of the most effective ways. BDQ is the first new anti-TB drug marketed in the past 50 years (3). It is highly potent *in vitro* and *in vivo* but has unacceptable cardiac adverse reactions. Therefore, appropriate modification of the structure of BDQ to determine a new generation of compounds with equivalent or better efficacy as BDQ, and the corresponding reduction of adverse reactions is undoubtedly a hot topic in this field. Herein, we reported WX-081, which achieves this goal by structural modification while retaining the central action group of BDQ.

WX-081 performed an excellent antimicrobial *in vitro* activity against both drug-susceptible and drug-resistant *M. tuberculosis* clinical isolates. The MIC of WX-081 against the five susceptible strains ranged from 0.117 to 0.219 $\mu$g/mL, and its activity was similar to that of BDQ, INH, RFP, and MFX and superior to that of KM. Moreover, WX-081 and BDQ achieved almost equivalent efficacies against 15 drug-resistant TB strains with the MIC in the range of 0.026 to 0.966 $\mu$g/mL. The results are consistent with previous studies (15, 16). On the other hand, WX-081 had some bactericidal activity but had a weak direct bactericidal effect on drug-resistant MTB, a similar effect as BDQ. In addition, time-kill curve analysis proved bactericidal activity of WX-081 in 14-day culture. The finding that CFU decreased also provided compelling evidence that WX-081 showed concentration-dependent characteristics of killing *M. tuberculosis* H37Rv and two clinical isolates. Besides, the effect of WX-081 was significant at the MBC concentration.

Based on the promising *in vitro* results, the drug was evaluated *in vivo*. WX-081 was found to have superior *in vivo* activity against *M. tuberculosis* in the acute infection model. WX-081 alone, at the minimum dose of 5 mg/kg, can already effectively reduce the bacterial load in the lungs. It decreased pulmonary bacilli loads 1000-fold compared with the vehicle group after a 20-day course of chemotherapy. Furthermore, daily administration of 20 mg/kg WX-081 monotherapy achieved all culturable organisms' clearance in mouse lungs in 20 days. The efficacy of WX-081 on acute infection models was similar to that of BDQ but significantly better than that of INH, and our data demonstrated a clear dose-response for WX-081 at three different doses, consistent with the bactericidal properties of BDQ (17–19). In the chronic infection model, WX-081 could produce a remarkable bacterial elimination in the lungs of mice at any dose in comparison to the vehicle group. With the increase of dose, the number of viable *M. tuberculosis* bacilli decreased more in mice.

Although WX-081 at the same dose caused more bacterial load in the lungs of mice than BDQ, the differences were not significant, suggesting that the efficacy of WX-081 was almost equivalent to that of BDQ. In addition, we observed that the activity of 5 mg/kg WX-081was superior in terms of reduction of CFU in the lungs of mice to that of 15 mg/kg RFP after an 8-week treatment duration. As a result, the potent bactericidal effect of WX-081 against sensitive and MDR-TB strains pointed out the enormous potential to treat mycobacterial infections.

WX-081 is a completely newly structured antituberculosis compound in the clinical stage. The preclinical PK study of WX-081 showed a molecule with good exposure, suitable elimination half-life, and much higher exposure in the lung as the target organ.

Interestingly, apart from the above advantages of WX-081, male and female beagle dogs hardly produced any significant adverse effects on the heart at daily doses for 14 consecutive days. Dysfunction of hERG easily leads to long QT syndrome and may increase the risk of sudden death in patients with cardiac ischemia (20). There were no qualitative ECG changes associated with the oral administration of WX-081 and may be associated with physiologically appropriate lengthening of the QT interval. However, several studies have reported QTc prolongation with BDQ (3, 21, 22). A meta-analysis of 1303 patients showed that the incidence of QT prolongation was 10.6% in clinical use, and 0.9% of patients discontinued taking BDQ due to prolongation of the QT

interval. Rouan et al. (17) indicated that elevated plasma concentrations of N-monodi-methyl metabolite (M2), a metabolite of BDQ, may lead to QT interval prolongation. Cardiac troponin I (cTnI) is one of the subunits of the troponin complex, a key component of cardiac muscle contraction, which can inhibit the ATPase activity of the actomyosin complex (23). There were no significant changes observed in the WX-081 group. Compared with the vehicle group, the concentration of cTnI of one male animal in the bedaquiline fumarate group was 440 pg/mL while that of all others was BLQ. Hence, WX-081, being structurally different from BDQ, may have a better safety profile, potentially further improving its clinical use. Even though no signal is found in the WX-081 preclinical tox study, the cardiac safety of WX-081 will be evaluated in clinical in further.

In conclusion, the new drug, WX-081, showed favorable activity against *M. tuberculosis* isolates and had good pharmacokinetic parameters, which could be considered a drug with potential antimicrobial activity similar to BDQ. More importantly, the ability to reduce cardiotoxicity exceeds that of BDQ, and patients may be able to obtain higher safety guarantees. In future studies, first, WX-081 and BDQ efficacy can be compared on more models such as C3HeB/FeJ mouse models. Irwin et al. (24) reported that BDQ alone exhibited decreased activity against *M. tuberculosis* in the lungs of C3HeB/FeJ mice compared with that in the lungs of BALB/c mice, accompanied by more BDQ-resistant mutants. This is associated with the production of hypoxia, cheese-like necrotizing granulomas in C3HeB/FeJ mice while BALB/c mice mainly form cellular, inflammatory lesions with intracellular bacteria (24). Second, a pharmacokinetic/pharmacodynamic model can be used to evaluate WX-081 by combining drug concentration with time, antibacterial effects, and, thus, clarifying the time process of antibacterial or bactericidal effect of WX-081 at a specific dose/concentration and a specific administration regimen. Third, it would also be of interest to assess if WX-081 had bactericidal and sterilizing activity attempting to obtain a drug targeting not only replicating bacteria but also nonreplicating population, which is predicted to significantly shorten treatment time (25). Finally, to prevent the emergence of drug resistance and maximize the efficacy of this potent new drug, the design, and implementation of combination regimens should be carefully considered. For example, PBTZ169, Decaprenyl-phosphoryl-ribose 2'-epimerase(DprE1) inhibitor, is an attractive candidate for the treatment of human tuberculosis (26). Makarov et al. (27) revealed that BDQ and PBTZ169 were synergistic against *M. tuberculosis* in a mouse model, which may be related to the fact that the weakening of the *M. tuberculosis* cell wall by PBTZ169 allows better penetration of BDQ into the cell and easier access to its target ATP synthase. Given the strong efficacy of WX-081, good PK, and no drug-related adverse cardiac reactions, WX-081 has the potential to be a next-generation anti-tuberculous drug and should be studied further to enable final clinical use to provide rich theoretical and data support.

## MATERIALS AND METHODS

**Compounds.** Isoniazid (INH), rifampicin (RFP), streptomycin (STR), levofloxacin (LVX), and ethambutol (EMB) were purchased from Sigma. Bedaquiline (BDQ) was offered by WuXi AppTec (Wuhan) Co., Ltd. and WX-081 was provided by WuXi AppTec (Shanghai) Co., Ltd.

**Strains.** Twenty clinical *M. tuberculosis* isolates were obtained from the National Clinical Laboratory on Tuberculosis, Beijing Chest Hospital. Among clinical *M. tuberculosis* isolates, five strains were fully drug-susceptible and 15 were drug-resistant, as determined using the microplate alamarBlue assay (MABA) (28). Drug resistance was defined as resistance to any of the following drugs: isoniazid (INH), rifampin (RFP), streptomycin (STR), levofloxacin (LVX), and ethambutol (EMB) using 1, 1, 2, 1, and 5 $\mu$g/mL, respectively, as the critical concentration. The *M. tuberculosis* were grown in Middlebrook 7H9 broth (Difco, USA) supplemented with 0.2% (vol/vol) glycerol, 0.05% Tween 80, and 10% (vol/vol) oleic acid-albumin-dextrose-catalase (OADC) (Becton, Dickinson, USA).

**MIC determinations.** WX-081 and BDQ MICs were determined by MABA, using 2-fold dilutions ranging from 0.4 to 0.00625 $\mu$g/mL (28). Briefly, bacteria (100 $\mu$L containing $2\times10^5$ CFU) were added to wells, yielding a final testing volume of 200 $\mu$L. The plates were incubated at 37°C. On day 7 of incubation, 12.5 $\mu$L of 20% Tween 80 and 20 $\mu$L of alamarBlue were added to all wells. After incubation at 37°C for another 24 h, the fluorescence was measured at an excitation wavelength of 530 nm and an emission wavelength of 590 nm. The MIC was defined as the lowest concentration eliciting a reduction in

fluorescence of ≥90% relative to the mean fluorescence of replicate drug-free controls. *M. tuberculosis* H37Rv was used as a drug-susceptible control.

**MBC determination.** CFU-based enumeration was utilized to determine minimal bactericidal concentrations (MBCs) against *M. tuberculosis* H37Rv and 5 *M. tuberculosis* clinical isolates in 96-well culture plates cultured after 7 days of incubation with the relative compounds. According to the CLSI guidelines, the MBC was determined by 99.9% killing of final inoculum (29). Initial CFU was calculated on the same day when culture is incubated in 96-well microplates. The culture concentration ranges were from MIC to 16× MIC for WX-081 and BDQ in MBC determination. After 7 days of incubation, culture aliquots from each well were plated on blank 7H10 agar plates, enriched with 10% OADC Enrichment (Difco, USA), and supplemented with ampicillin (50 $\mu$g/mL), polymyxin B (33.3 $\mu$g/mL), trimethoprim (20 $\mu$g/mL) and cycloheximide (200 $\mu$g/mL) to prevent contamination of other bacteria. Besides, due to potential drug carryover issues with WX-081 and BDQ, 0.4% (weight/volume) activated charcoal should be added to the 7H10 solid media to act as an adsorbent (30). CFU of all plates was counted after 28 days in an incubator. The MBC was defined as the lowest concentration of the drugs that had effect can be seen by at least a 3 $\log_{10}$ decrease in CFU compared to the initial CFU.

**Time-kill curve.** Time-kill curve analyses were performed by culturing *M. tuberculosis* in a 7H10 solid agar medium. Bacterial culture ($2 \times 10^5$ CFU/mL) was added to wells, in the presence of WX-081 and BDQ concentrations in doubling dilutions ranging from 0.5× MIC to 16× MIC in 24-well plates. For all strains, MICs were determined previously. The *M. tuberculosis* simultaneously was cultured in drug-free 7H9 broth served as growth control for each strain. Ten-fold serial dilutions of culture suspension were extracted at 0, 3, 7, 10, and 14 days and plated on a 7H10 solid medium. After 28 days of incubation, CFU counts of all the plates were recorded. The killing rate of *M. tuberculosis* by each drug could be determined by plotting $\log_{10}$ CFU/mL versus culture time.

**Pharmacokinetic studies in animals.** The BALB/c mice, rats, and beagle dogs were used in the PK study and fasted overnight before dosing. Animal doses were administered in two ways. One is oral administration, configured to form a 6.25 mg/kg WX-081, with 0.5% carboxymethylcellulose (CMC), the other is the WX-081 with an intravenous injection of 1.0 mg/kg (formulated in 40% hydroxypropyl-$\beta$-cyclodextrin and 50 mM citrate buffer at pH 3). Plasma samples were collected from three animals at each time point. Plasma concentrations were determined by liquid chromatography-tandem mass spectrometry (LC-MS/MS). The bioanalytical method by LC-MS/MS for pharmacokinetic (PK) studies in animal and toxicokinetic (TK) in tox studies have been validated. The total area under the concentration-time curve ($AUC_{0-inf}$), the elimination half-time ($t_{1/2}$), the peak concentration ($C_{max}$), and the time to reach peak concentration ($T_{max}$) of samples were determined directly from the experimental data using WinNonlin V6.2.1.

**Aerosol infection with *M. tuberculosis*.** Specific pathogen-free (SPF) BALB/c male mice (18 to 20 g) were infected via aerosol with *M. tuberculosis* H37Rv using a Glas-Col inhalation system. The acute infection model used a high-dose aerosol infection with a suspension of $5 \times 10^6$ CFU/mL to deposit 50 to 100 bacilli into the lungs of each mouse, as previously described (31). The chronic infection model used a low-dose aerosol infection with a suspension of $2 \times 10^6$ CFU/mL to deposit 20 to 50 bacilli into the lungs. Five mice (acute and chronic model) were humanely killed 3 days after infection and on the day of treatment initiation (D0) to determine the number of bacteria implanted in the lungs and at the start of treatment, respectively.

**Chemotherapy.** Treatment was initiated 10 days after infection in the acute model, and 28 days after infection in the chronic model. The drugs and compounds were prepared weekly by suspension in 0.5% (wt/vol) CMC. Groups of 10 mice in the acute model and 8 mice in the chronic model were dosed for 5 consecutive days each week.

In the acute model, the vehicle control mice received an equal volume of 0.5% CMC and the positive-control mice received INH (25 mg/kg) or BDQ (5, 10, or 20 mg/kg). The test mice received WX-081 (5, 10, or 20 mg/kg) treatment was administered for up to 3 weeks.

In the chronic model, the vehicle control mice received CMC and the positive-control mice received RIF (10 mg/kg) and BDQ (5, 10, or 20 mg/kg). The test mice received WX-081 at 5, 10, or 20 mg/kg. The treatment duration was 8 weeks.

**Assessment of treatment efficacy.** Efficacy was assessed based on the lung CFU counts at selected time points during treatment. Mice were sacrificed 3 days after the last day of treatment to reduce carryover effects. Quantitative cultures of lung homogenates were performed on 0.4% charcoal-containing selective 7H11 plates, to minimize carryover effects (32). The plates were incubated for up to 6 weeks at 37°C before the final CFU counts were determined.

**Toxicology studies in a dog.** One head-to-head tox study of WX-081 and bedaquiline fumarate was performed at WuXi AppTec (Suzhou) Co., Ltd. (testing facility) and conducted in compliance with the most recent version of the good laboratory practice (GLP) regulations. All applicable portions of the study conformed to the regulations and guidelines regarding animal care and welfare as AAALAC. Based on the above, the study was designed for one 2-week high dose at 200 mg/kg head-to-head comparison study of WX-081 with BDQ in a dog (*n* = 3 dog/sex/group). A human ether-á-go-go-related (*hERG*) ion channel *in vitro* assay was performed in Shanghai Institute of Materia Medica, Shanghai, for the evaluation of cardiac liability testing. The assay was a manual patch-clamp method using electrophysiology in an hERG-CHO cell.

**Statistical analysis.** Lung CFU counts were log-transformed before analysis, and mean CFU counts were compared by one-way analysis of variance with Dunnett's posttest to control for multiple comparisons. The Mann-Whitney test was used to test for significance on nonnormally distributed CFU data. All analyses were performed with GraphPad Prism version 5 (GraphPad, San Diego, CA). A *P* value of 0.05 was considered significant.

## ACKNOWLEDGMENTS

We thank Shitong Huan, who is a Senior Program Officer of Health, China Program Bill and Melinda Gates Foundation, Khisi Mdluli, and Marja van Zeijl, who belong to Senior Program officer HIV, and TB Drugs Bill and Melinda Gates Foundation for reviewing the manuscript and making pertinent suggestions to us.

This work was supported by the Beijing Hospitals Authority Clinical Medicine Development of Special Funding Support (ZYLX202123). We declare no conflict of interest.

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
