## [Reviewer comments · Microbiology Spectrum]

Microbiology Spectrum

Sudapyridine (WX-081) , a novel compound against Mycobacterium tuberculosis

Yu Lu, Rong Yao, Bin Wang, Lei Fu, Lei Li, Kejun You, and Yongguo Li

Corresponding Author(s): Yu Lu, Beijing Tuberculosis and Thoracic Tumor Research Institute; Beijing Chest Hospital, Capital Medical University

Review Timeline:

Submission Date:	December 10, 2021
Editorial Decision:	December 21, 2021
Revision Received:	January 13, 2022
Accepted:	January 16, 2022

Editor: Luiz Pedro de Carvalho

Reviewer(s): The reviewers have opted to remain anonymous.

Transaction Report:

DOI: <https://doi.org/10.1128/spectrum.02477-21>

December 21, 2021

Dr. Yu Lu
Beijing Tuberculosis and Thoracic Tumor Research Institute; Beijing Chest Hospital, Capital Medical University
Pharmacology
Beijing Chest Hospital
Capital Medical University
Beijing
China

Re: Spectrum02477-21 (*Sudapyridine (WX-081)* , a novel compound against *Mycobacterium tuberculosis*)

Dear Dr. Yu Lu:

Please, review English usage throughout the manuscript and include a figure/scheme with the structure of sudapyridine, made with a professional drawing software, such as chemdraw or equivalent.

Link Not Available

Sincerely,

Luiz Pedro de Carvalho

Journals Department
Reviewer comments:

Staff Comments:

Preparing Revision Guidelines

- Point-by-point responses to the issues raised by the reviewers in a file named "Response to Reviewers," NOT IN YOUR

COVER LETTER.

- Upload a compare copy of the manuscript (without figures) as a "Marked-Up Manuscript" file.
- Each figure must be uploaded as a separate file, and any multipanel figures must be assembled into one file.
- Manuscript: A .DOC version of the revised manuscript
- Figures: Editable, high-resolution, individual figure files are required at revision, TIFF or EPS files are preferred

Please return the manuscript within 60 days; if you cannot complete the modification within this time period, please contact me. If you do not wish to modify the manuscript and prefer to submit it to another journal, please notify me of your decision immediately so that the manuscript may be formally withdrawn from consideration by Microbiology Spectrum.

January 16, 2022

Dr. Yu Lu
Beijing Tuberculosis and Thoracic Tumor Research Institute; Beijing Chest Hospital, Capital Medical University
Pharmacology
Beijing Chest Hospital
Capital Medical University
Beijing
China

Re: Spectrum02477-21R1 (*Sudapyridine (WX-081)* , a novel compound against *Mycobacterium tuberculosis*)

Dear Dr. Yu Lu:

Your manuscript has been accepted, and I am forwarding it to the ASM Journals Department for publication. You will be notified when your proofs are ready to be viewed.

Finally, you might want to abbreviate Rifampicin to RIF, instead of RMP or RFP. You could amend that during the galley proof stage.

Sincerely,

Luiz Pedro Carvalho
Editor, Microbiology Spectrum
